# COFFEE: Counterfactual Fairness for Personalized Text Generation in Explainable Recommendation

**Nan Wang[1,3]***, **Qifan Wang[2], Yi-Chia Wang[2], Maziar Sanjabi[2], Jingzhou Liu[2]**
**Hamed Firooz[2], Hongning Wang[3], Shaoliang Nie[2]**
[1]Netflix Inc., Los Gatos, California, USA
[2]Meta AI, Menlo Park, CA, USA
[3]University of Virginia, VA, USA

## Abstract

As language models become increasingly integrated into our digital lives, Personalized Text Generation (PTG) has emerged as a pivotal component with a wide range of applications. However, the bias inherent in user written text, often used for PTG model training, can inadvertently *associate different levels of linguistic quality with users' protected attributes*. The model can inherit the bias and perpetuate inequality in generating text w.r.t. users' protected attributes, leading to unfair treatment when serving users. In this work, we investigate fairness of PTG in the context of personalized explanation generation for recommendations. We first discuss the biases in generated explanations and their fairness implications. To promote fairness, we introduce a general framework to achieve *measure-specific counterfactual fairness* in explanation generation. Extensive experiments and human evaluations demonstrate the effectiveness of our method.

## 1 Introduction

Personalized text generation (PTG) has extensive applications, such as explainable recommendation (Zhang and Chen, 2020; Chen et al., 2021), post generation (Yuan and Huang, 2019; He et al., 2021), and conversational systems (Zhang et al., 2018, 2019; Lee et al., 2021). The auto-generated text, functioning at the frontier of human-machine interaction, influences users' decisions and transforms their way of thinking and behaving. However, due to its immense power and wide reach, PTG can inadvertently give rise to fairness issues and lead to unintended consequences (Alim et al., 2016; Bordia and Bowman, 2019; Blodgett et al., 2020).

In this work, we investigate the fairness issues in PTG, focusing on one of the mostly studied settings: generating natural language explanations for

---
* The research was conducted when the author was an intern at Meta and a PhD candidate at the University of Virginia. Correspondence to nanw@netflix.com

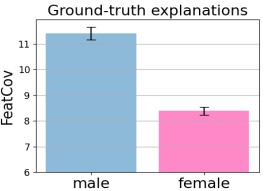 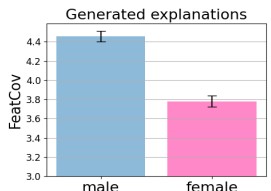

Figure 1: Left: average FeatCov of ground-truth explanations in the train set. Right: average FeatCov of explanations generated on the test set by PETER.

recommendations (Wang et al., 2018; Chen et al., 2021; Yang et al., 2021; Li et al., 2021a; Yang et al., 2022). Personalized explanation generation aims to provide a user with descriptive paragraphs on recommended items that align with his/her preferences, enabling more informative and accurate decision-making. The generators are typically language models trained on user written reviews from e-commerce platforms (Zhang et al., 2018; Ni et al., 2019; Yang et al., 2021), where sentences related to item descriptions are retained to construct the ground-truth explanations. However, due to historical, social, or behavioral reasons, inherent bias may exist within the review text, associating specific linguistic characteristics with the users' protected attributes such as gender or race (Newman et al., 2008; Alim et al., 2016; Volz et al., 2020). While certain linguistic features that capture the diversity of language use (Newman et al., 2008; Groenwold et al., 2020) are suitable for personalization, others pertaining to the *linguistic quality* of explanations, such as informativeness or detailedness, (Louis, 2013), should be excluded. Failure to do so can result in unfair treatment when serving users.

As an example, we investigate the explanation generation on Amazon Movies[1] with the personalized transformer model PETER (Li et al., 2021a). We adopt feature coverage (FeatCov, the number of unique features mentioned about a movie) as an automatically measurable metric of explanation

---
[1]The details about experiment setup is in Section 6.

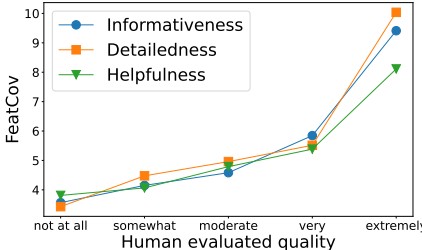

Figure 2: FeatCov vs. human evaluated quality measures. Each measure is rated in a five-category scale. $p$-valus $< 0.05$ in Kruskal-Wallis H test for all three criteria (see Appendix D for detailed test results).

quality. As shown in Figure 1 (left), reviews from male users generally have higher FeatCov on the target movies than those from female users (Fan-Osuala, 2023)[2]. A model trained on such data can inherit the bias and generate explanations discriminately when serving users—higher FeatCov for males than females, as indicated in Figure 1 (right). To further substantiate the bias issue, we conducted a user study and found strong positive correlation, shown in Figure 2, between FeatCov and human evaluated quality criteria including informativeness, detailedness and helpfulness. The observations remain consistent when results from male and female human evaluators are analyzed separately, affirming that both genders consider FeatCov a significant indicator of quality. This study demonstrates the bias in FeatCov as a concerning issue for PTG. Details of the user study are in Section 7.

In particular, the bias observed in training data originates from various factors such as different demographic behavior patterns that are beyond our control. But the system should not inherit the bias and act discriminately by generating explanations of different quality for different users—the lack of informative reviews from users of a demographic group should not prevent them from receiving informative explanations. Without proper intervention, the system can exhibit such bias, thus adversarially affecting the users' experience, reliance and trust in the system (Tintarev and Masthoff, 2015). From a broader perspective, the bias can reinforce itself by influencing how users write online, and jeopardize fairness in the long run (Schwartz et al., 2016; Alim et al., 2016; Bordia and Bowman, 2019).

To mitigate the issue, we take a causal perspective to dissect the bias and enforce counterfactual fairness (CF) (Kusner et al., 2017) in personalized explanation generation: the quality of generated

explanations should not differentiate a user in the real world vs. in the counterfactual world where *only* the user's protected attribute (e.g., gender) is changed. This problem is essential and unique compared to fairness problems in the literature (Section 2), and imposes specific technical challenges (Section 4). To achieve the goal, we develop a general framework, COFFEE, for **CO**unter**F**actual **F**airn**E**ss in **E**xplanation generation. COFFEE treats a user's protected attribute value as a separate token input to the model, and disentangles its representation from the user's representation (Ma et al., 2019; Locatello et al., 2019b; Zheng et al., 2021). By controlling the input attribute values for counterfactual inference (CI), we impose a measure-specific CF constraint (Russell et al., 2017) on generated explanations. Then a novel fair policy learning scheme is developed to optimize CF with carefully designed rewards, which generalizes to any single or combination of quality measures. We use user-side fairness by default in discussions, but COFFEE is general to ensure fairness on either user or item side in the two-sided market (Wang and Joachims, 2021), and be adapted to different models. Extensive experiments and rigorous user studies demonstrate COFFEE's superiority in achieving fairness and maintain high generation performance compared to baselines.

## 2 Background

**Uniqueness and Importance:** Fairness in machine learning (ML) is originally studied in automatic decision-making systems that directly impose "significant" or "legal" effects on individuals (Voigt and Bussche, 2017). Such fairness considerations often revolve around *resource allocation* by ML models, exemplified in contexts like loan assessments (Lee and Floridi, 2021) or job applications (Singh and Joachims, 2018), where model predictions can unfavorably affect a protected group (Du et al., 2021; Mehrabi et al., 2021).

In contrast to resource allocation fairness, NLP researchers primarily examine the *representational fairness* (Blodgett et al., 2020; Liang et al., 2021; Sheng et al., 2021) regarding how language models shape social biases and stereotypes through natural language understanding (NLU) or generation (NLG). In the realm of NLG, fairness particularly concerns how generated text may contain biased information about a specific demographic group. For instance, Huang et al. (2020) analyze the sentence completion by a GPT-2 model, and find dif-

---

[2]We use binary gender for case study, but our work generalizes to any binary or non-binary attributes.

ferent sentiment distributions of completed sentences when the occupation word is counterfactually changed in the prompts. Bordia and Bowman (2019) revealed a more frequent co-occurrence of the 'doctor' with male pronouns and 'nurse' with female pronouns in generated text. However, these biases, directly encapsulated within the text, can be more easily analyzed. To the best of our knowledge, our work is pioneering in exploring how personalization, bridging NLP and recommendation, associates the bias in NLG with protected attributes of users.

**Fairness Notions:** Various fairness notions exist in the literature, with group-wise fairness notions being the firstly studied ones (Zafar et al., 2015; Hardt et al., 2016; Zafar et al., 2017). Yet, group-wise fairness has different quantitative definitions that are generally incompatible (Kleinberg et al., 2017; Berk et al., 2021). Some definitions can even exacerbate discrimination (Kusner et al., 2017). Individual fairness (Zemel et al., 2013; Joseph et al., 2016) requires similar users to receive similar predictions. But it relies on carefully chosen domain-specific similarity metrics (Dwork et al., 2012). In contrast, counterfactual fairness (CF) (Kusner et al., 2017), considering fairness from a causal perspective, has gained prominence recently as a more robust fairness notion (Russell et al., 2017; Wu et al., 2019; Makhlouf et al., 2020), which can also enhance group-wise fairness in certain scenarios (Zhang and Bareinboim, 2018; Khademi et al., 2019).

Though CF has been studied in some non-personalized NLP tasks (Huang et al., 2020; Garg et al., 2019), most existing works study the dependency of model outputs on attribute-specific words within the input text (Blodgett et al., 2020; Liang et al., 2021; Sheng et al., 2021). In such cases, CI can be easily performed on the input text itself, such as changing male pronouns to female pronouns (Huang et al., 2020; Garg et al., 2019). However, CF in PTG necessitates CI on the protected attributes of users being served-an area yet to be thoroughly explored.

## 3   Problem Formulation

In the following discussions, we consider a single protected attribute on the user side for simplicity, but our proposed framework is versatile to accommodate multiple attributes on either the user or the item side. The value of a user's protected attribute is denoted by a variable $A \in \mathcal{A}$,

where $\mathcal{A}$ is the set of possible attribute values, e.g., $\mathcal{A} = \{male, female, other\}$ for gender. Each dataset entry is a tuple of $(u, i, a, e)$, corresponding to user ID, item ID, observed attribute value, and ground-truth explanation. The explanation generator $G_\theta$ is a language model parameterized by $\theta$. Given a user $u$, an item $i$, and observed attribute value $a$, an explanation can be sampled from the generator as $Y \sim G_\theta(u, i | A = a)$. The linguistic quality of any explanation $y$ is measured by a function $Q(y)$. Notably, we treat $Q$ as a *black box oracle*—a quality measure that can only be queried. This is essential in practice and offers the flexibility to arbitrarily tailor $Q$ based on the fairness requirements of the application. An explanation can be gauged in various ways by customizing $Q$, such as using an explicit function, human evaluation, or a tool provided by authorities. We assume, without loss of generality, that higher $Q$ values represent superior quality. CF on any measure $Q$ of explanations is achieved when, given a user $u$ and an item $i$,

$$P(Q(Y_{A\leftarrow a})|u, i, a) = P(Q(Y_{A\leftarrow a'})|u, i, a), \quad (1)$$

where $Y_{A\leftarrow a'} \sim G_\theta(u, i | A = a')$ is the explanation generated when we counterfactually assign the value of the user's protected attribute by $A \leftarrow a', a' \neq a$. The right side of Eq. (1) evaluates the quality distribution of explanations generated *had the user's protected attribute value been $a'$*, given that the *observed attribute value is $a$* (Kusner et al., 2017; Li et al., 2021b).

Denote the loss of the generator for a given user $u$ and item $i$ by $\mathcal{L}_{gen}(G_\theta(u, i | A = a), e)$, which is typically the negative log-likelihood (NLL) loss or a combination of several losses (Li et al., 2017; Yang et al., 2021; Li et al., 2021a). We consider training the generator for fair explanation generation as a constrained optimization problem:

$$\begin{aligned} \min &\mathcal{L}_{gen}(G_\theta(u, i | A = a), e) \\ s.t. \ &\mathbb{E}_{Y_{A\leftarrow a}}[Q(Y_{A\leftarrow a})|u, i, a] = \quad (2) \\ &\mathbb{E}_{Y_{A\leftarrow a'}}[Q(Y_{A\leftarrow a'})|u, i, a] \end{aligned}$$

For ease of presentation, we consider a single user-item pair, and the total loss on a dataset is simply summed over all user-item pairs with the constraint applied to every pair. In this work, we apply the first-order moment of the quality of generated explanations to construct the constraint, and leave the extension to other moment-matching constraints for future work. We further simplify the expression of the constraint as $\mathbb{E}[Q(Y_{A\leftarrow a})] = \mathbb{E}[Q(Y_{A\leftarrow a'})]$.

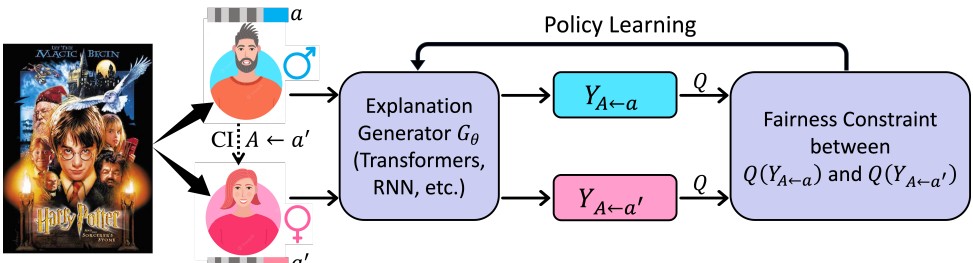

Figure 3: Given a user with attribute value $a$ and a recommended item, COFFEE performs CI by switching the attribute value to $a'$ to get the counterfactual user, which is achieved by disentangled attribute embeddings. Explanations $Y$ for both the real user and the counterfactual user are sampled from the generator, evaluated by $Q$. COFFEE then updates the generator's parameters $\theta$ by policy learning from the fairness constraint.

## 4 COFFEE

User and item IDs are conventionally input to the generator for explanation generation, represented as learned embedding vectors (Li et al., 2017; Wang et al., 2018; Li et al., 2021a). However, the user's protected attribute, entangled with their preference, is implicitly encoded in the representations (Locatello et al., 2019a), hindering its direct manipulation for CI. One way explored for CF in personalization involves removing all information about the protected attribute in representations via discriminators (Zemel et al., 2013; Li et al., 2021b; Wang et al., 2022). Although this improves fairness, it detrimentally affects personalization and eliminates desired characteristics linked to the protected attribute. In contrast, we aim to enforce the independence of the protected attribute from *any specified quality measure $Q$* on generated explanations, while preserving explanation content-dependence to sustain high personalization performance.

To enable CI, COFFEE considers a user's protected attribute value as a separate token input to the model, along with user and item IDs, and learns disentangled attribute representations from the user representation to encapsulate the effect of each attribute value in explanation generation (Locatello et al., 2019a,b; Ma et al., 2019; Zheng et al., 2021). This mirrors methods in controllable text generation (Oraby et al., 2018; Shu et al., 2020; Dathathri et al., 2020), where disentangled attribute tokens are utilized as inputs to modulate the topic, sentiment, or style of generated text. CI is then achieved by altering the attribute token input to the model. Subsequently, we enforce a fairness constraint based on the explanations generated pre and post CI, and establish a policy learning method for optimizing this constraint. An illustration of the COFFEE framework is shown in Figure 3.

### 4.1 Disentangled Attribute Representation

For a given tuple $(u, i, a)$, we denote the representation for the attribute value $a$ as $\mathbf{r}_a$, the user's preference representation (independent from the protected attribute) as $\mathbf{r}_u$ and item representation as $\mathbf{r}_i$. The complete user representation is $\mathbf{r}_u^a = [\mathbf{r}_a, \mathbf{r}_u]$. Correspondingly, when performing $A \leftarrow a'$ on user $u$, we change the input attribute token from $a$ to $a'$, and the new user representation becomes $\mathbf{r}_u^{A \leftarrow a'} = [\mathbf{r}'_a, \mathbf{r}_u]$. Note that each attribute value has its own representation, and is shared across all users having that same attribute value. For instance, all male users' attribute representation is the same vector $\mathbf{r}_{male}$. We can do the same for item-side attributes as $\mathbf{r}_i^a = [\mathbf{r}_a, \mathbf{r}_i]$.[3]

Simply separating the user's protected attribute and preference representations does not guarantee that $\mathbf{r}_u$ will not contain any information about the protected attribute, inhibiting the accuracy of CI. To further enforce the disentanglement, we introduce a discriminator $D(\mathbf{r}_u)$, and add an adversarial loss on $\mathbf{r}_u$ in Eq. (2) as

$$\min \ \mathcal{L}_{gen}(G_\theta(u, i | A = a), e) + \lambda_D \log(D(r_u, a))$$
$$s.t. \ \mathbb{E}[Q(Y_{A \leftarrow a})] = \mathbb{E}[Q(Y_{A \leftarrow a'})], \forall a' \in \mathcal{A}, a' \neq a, \quad (3)$$

where $D(r_u, a)$ is the probability of predicting the correct attribute value $a$. In this way, we adversarially remove the protected attribute information from $\mathbf{r}_u$, and enforce $\mathbf{r}_a$ to capture all the attribute information. During mini-batch optimization, we alternate between the parameter updates of the model and the discriminator as follows: (1) $X$ batches of updates minimizing the loss Eq. (3) with $D$ fixed, and (2) $Z$ batches of updates maximizing the the loss Eq. (3) with the generator $G_\theta$ fixed.

---

[3]We can introduce $K \geq 2$ attribute tokens, each mapped to its disentangled representations. Sum instead of concatenation of embeddings can be used when $K$ is large.

## 4.2 Policy Learning for Fairness

Once the user, item, and attribute representations are *learned and fixed*, we can optimize the generator w.r.t. the CF constraint. Due to the non-convexity of the constrained fairness optimization problem, closed-form solutions are unattainable. We add the constraint with Lagrangian relaxation as a regularization in the loss (Russell et al., 2017)

$$\min \mathcal{L}_{all} + \lambda \big| \mathbb{E}[Q(Y_{A \leftarrow a})] - \mathbb{E}[Q(Y_{A \leftarrow a'})] \big|, \quad (4)$$

with $\mathcal{L}_{all} = \mathcal{L}_{gen}(G_\theta(u, i | A = a), e) + \lambda_D \log(D(r_u, a))$ in Eq. (3) denotes all other losses except for the CF constraint, and $\lambda$ is a hyperparameter for fairness-utility trade-off.

However, standard gradient methods cannot be directly applied to optimize the constraint: explanations are discretely sampled from the generator, and $Q$ is an oracle. Instead, we consider policy learning for fairness optimization, where the explanation distribution imposed by the generator is considered as the policy for explanation generation, and sampled explanations are actions. Concretely, for estimating the expectations in the regularization, we sample $N$ explanations and calculate the average quality of explanations sampled in both the real-world and the counterfactual world. Denote the regularization term as $\mathcal{L}_{fair}$, the expectations in the regularization can be estimated as:

$$\mathcal{L}_{fair} = \big| \mathbb{E}[Q(Y_{A \leftarrow a})] - \mathbb{E}[Q(Y_{A \leftarrow a'})] \big|$$
$$\approx \text{sign}(\Delta) \left( \frac{1}{N} \sum_{k=1}^{N} Q(y_{A \leftarrow a}^k) - \frac{1}{N} \sum_{k=1}^{N} Q(y_{A \leftarrow a'}^k) \right), \quad (5)$$

$\Delta = \frac{1}{N} \sum_{k=1}^{N} Q(y_{A \leftarrow a}^k) - \frac{1}{N} \sum_{k=1}^{N} Q(y_{A \leftarrow a'}^k)$. For an explanation $y_{A \leftarrow a}^k$ sampled in the real-world, its contribution to the unfairness in regularization term is thus $\frac{1}{N}\text{sign}(\Delta)Q(y_{A \leftarrow a}^k)$. To improve fairness by minimizing the regularization, the reward for $y_{A \leftarrow a}^k$ is considered as $r(y_{A \leftarrow a}^k) = -\frac{1}{N}\text{sign}(\Delta)Q(y_{A \leftarrow a}^k)$. Similarly, the reward for a sampled explanation $y_{A \leftarrow a'}^k$ in the counterfactual world is $r(y_{A \leftarrow a'}^k) = \frac{1}{N}\text{sign}(\Delta)Q(y_{A \leftarrow a'}^k)$. In this way, we convert the CF optimization into maximizing the rewards of generated explanations.

Although the rewards are designed to minimize the difference in the qualities of explanations from the real vs. counterfactual world, we lack direct control over how the difference should be minimized during optimization. The optimization may arbitrarily improve or decrease the quality of explanations to achieve fairness, e.g., always generating low quality explanations, which greatly hurt

their utility in practice. To address this, we further design a weighting mechanism to calibrate the rewards such that the optimization focuses more on improving (or decreasing) the quality measure for an attribute value for achieving fairness. This empowers the designer to better control the optimization and the utility-fairness trade-off. Specifically, we introduce a quality promotion weight $\eta \in [0, 1]$ to re-weigh the rewards of explanations in the world with lower expected quality, and use $1 - \eta$ to reweigh the rewards of explanations in the other world. The resulting calibrated rewards are:

$$r_w(y_{A \leftarrow a}^k) = -\frac{1}{N}\text{sign}(\Delta)Q(y_{A \leftarrow a}^k) \cdot w(\Delta)$$
$$r_w(y_{A \leftarrow a'}^k) = \frac{1}{N}\text{sign}(\Delta)Q(y_{A \leftarrow a'}^k) \cdot (1 - w(\Delta))$$

where $w = \frac{\text{sign}(\Delta)+1}{2}(1 - \eta) + \left(1 - \frac{\text{sign}(\Delta)+1}{2}\right)\eta$. We can leverage this weight to guide the fairness optimization: if $\eta > 0.5$, the algorithm will focus more on improving the quality of low-quality explanations for fairness. Finally, for stability and faster convergence, we apply the advantage trick (Mnih et al., 2016), and each reward is its difference from the average reward in its corresponding world:

$$r_{adv}(y_{A \leftarrow a}^k) = r_w(y_{A \leftarrow a}^k) - \bar{r}_w(y_{A \leftarrow a}^k)$$
$$r_{adv}(y_{A \leftarrow a'}^k) = r_w(y_{A \leftarrow a'}^k) - \bar{r}_w(y_{A \leftarrow a'}^k) \quad (6)$$

The policy gradient (Sutton et al., 1999; Yang et al., 2021) on $\theta$ can then be estimated as:

$$\nabla_\theta \mathcal{L}_{fair} \approx \sum_{k=1}^{N} \nabla_\theta \log G_\theta(y_{A \leftarrow a}^k) r_{adv}(y_{A \leftarrow a}^k)$$
$$+ \sum_{k=1}^{N} \nabla_\theta \log G_\theta(y_{A \leftarrow a'}^k) r_{adv}(y_{A \leftarrow a'}^k) \quad (7)$$

Intuitively, during optimization, the probability of sampled explanations that lead to the unfairness will be demoted, while the probability of those that contribute to fairness will be promoted.

To train a COFFEE model, we first pre-train the model without the fairness constraint. Then we fix the latent representations of users, items and attributes, and add the fairness constraint in the objective function for fine-tuning the model.

## 5 Applying COFFEE to Existing Models

Existing models for personalized explanation generation mostly adopt either Transformer (Li et al., 2021a) or RNN (Li et al., 2017; Chen et al., 2021; Yang et al., 2021). In this section, we demonstrate the application of COFFEE to PETER (Li

Table 1: Statistics of the datasets and the protected attributes used in experiments. The candidate values of the attributes are indicated in parentheses.

| | #Users | #Items | #Reviews | Attribute Values |
|---|---|---|---|---|
| Video Games | 9,511 | 5,173 | 59,374 | male, female |
| Movies & TV | 28,001 | 12,888 | 265,646 | male, female |
| Yelp | 35,714 | 24,900 | 1,499,291 | $, $$, $$$ |

et al., 2021a), which is an explanation generation model based on personalized Transformers. In Appendix A, we also illustrate the application of COFFEE to NRT (Li et al., 2017), which is based on RNN. However, it is worth noting that COFFEE is a general framework that can be applied to most of existing explanation generation models, as long as the model is based on user/item representation learning and can be fine-tuned.

In PETER (Li et al., 2021a), user and item IDs, represented as special tokens, are appended to the start of each explanation sequence before being inputted to transformer layers. A context prediction loss is introduced for enhanced personalization in addition to the NLL loss, which predicts the words in the explanations regardless of their order. We denote the overall loss as $\mathcal{L}_{peter}$, which is the sum of the NLL loss and context prediction loss (Li et al., 2021a). To apply COFFEE to PETER, we introduce an attribute token at the start of each sequence for disentangled attribute embedding. The discriminator is applied at the input embedding layer to enhance disentanglement. User, item, and attribute tokens can attend to each other, while the explanation words can only attend to past tokens. When applying COFFEE to PETER, the loss is constructed by replacing $\mathcal{L}_{gen}$ in Eq. (3) by $\mathcal{L}_{peter}$.

## 6 Experiments

We conduct experiments on three public review datasets: Amazon Video Games, Amazon Movies & TV, and Yelp Restaurant, and compare COFFEE with multiple baseline models. Due to space limit, we present the results based on PETER, and put example explanations and results based on NRT in Appendix E.

### 6.1 Experiment Setup

**Datasets.** In Amazon Games and Amazon Movies review datasets[4], we use users' binary gender (male, female) as the protected attribute. We study the fairness on the item side on the Yelp dataset[5],

[4]https://nijianmo.github.io/amazon/index.html
[5]https://www.yelp.com/dataset

where a restaurant's price range ($, $$, $$$) is used as the protected attribute, as the system may not discriminate a restaurant simply because the restaurant owner sets lower prices for equally good food and service than its counterparts. The statistics of the datasets are summarized in Table 1. The details of data processing are described in Appendix B.

On both Amazon Games and Movies, the raw model tends to generate higher FeatCov explanations for male users than female users, aligned with the existing findings that male users usually write more detailed reviews (Fan-Osuala, 2023). On Yelp, the model generates more thorough feature descriptions for pricier restaurants than their less expensive counterparts, as shown in Table 4 in Appendix B. This could be a reflection of users' heightened attention to experiences in more expensive restaurants due to higher expenditure. These observations also align with the biases present in the respective training datasets.

**Baselines.** As we are the first to study the quality fairness in PTG, there is no existing work that directly addresses this problem. We adapt popular methods in the fairness literature for the purpose. Below we briefly introduce them and describe their detailed implementations in Appendix B.

• RAW: Original PETER model without modification, which is also the base for other baselines.
• ADV (in-processing): Adversarially removing the sensitive information in user or item representations by adding a discriminator (Li et al., 2021b).
• NORM (pre-processing): Normalizing the training data to remove the bias on group-level.
• BT (pre-processing): Back-translation has been shown to help normalize text and reduce bias (Rabinovich et al., 2017; Christiansen et al., 2021). We pre-process the training data by translating the explanations to Chinese and then back to English.
• ATTR: To evaluate the effectiveness of the disentanglement and optimizing the fairness constraint in COFFEE, we use the model trained without the constraint (equivalent to $\lambda = 0$ in COFFEE) but with disentangled attribute representations.
• NATTR (post-processing): We disable the protected attribute token in ATTR for generating explanations during inference.

We train each model on the training set, tune hyper-parameters on the validation set, and report the results on the testing set. Each reported result point is averaged over 3 runs. We put the detailed model specifications and parameter tuning in Ap-

pendix C.

**Metrics.** Our evaluation consists of two parts: fairness and utility. Fairness requirements are subjective, contingent on both the application and the the designer's objective (Hu and Chen, 2020; Khademi et al., 2019). COFFEE is designed to accommodate different fairness specifications by appropriately defining $Q$. To select meaningful quality evaluation metrics in experiments, we evaluated explanations with different measures such as FeatCov, length, grammar, redundancy, structure (Zhu and Bhat, 2020). We found that only FeatCov consistently shows significant correlations to human perceptions of explanation quality such as informativeness, detailedness, and helpfulness. The detailed correlation results are shown in Table 6 in Appendix D. Therefore, we specify the quality measure function in COFFEE as $Q_{feat}$, which directly corresponds to an explanation's FeatCov. Furthermore, experiment findings suggest a substantial influence of the number of tokens in explanations on FeatCov. Hence, we additionally explore a combination of FeatCov and number of tokens to facilitate the fairness optimization on FeatCov, specifying $Q$ as $Q_{feat} + Q_{numtoken}$. We refer to this variant, with $Q$ augmented by the number of tokens, as COFFEE-NT. This modification also helps validate COFFEE's ability to adapt to different quality measures in optimizing fairness.

• Fairness metrics. We adopt both individual and group-wise fairness metrics evaluated on FeatCov, with and without counterfactual perspectives. Detailed metric formulas are in Appendix B. The first metric, Ind-CF, assesses individual CF as defined by (Kusner et al., 2017), which is equivalent to the regularization term in Eq. (4) averaged over all test set user-item pairs. We also evaluate the counterfactual effects on group-level (Coston et al., 2020). For a demographic group with attribute value $a$, we counterfactually alter all group members' attribute value to $a' \neq a$, and compute the change of the average FeatCov of the group. The average change is denoted as Grp-CF. Additionally, we evaluate COFFEE's generalization to improve group-wise fairness evaluated by demographic disparity (DDP) (Zafar et al., 2015). Metrics based on CF can only be evaluated on baselines that feature disentangled attribute representations, such as ATTR and COFFEE. All fairness metrics are *lower the better*.

• Utility metrics. Besides commonly used BLEU-{1,4} (Papineni et al., 2002) and ROUGE-{1,2,L}

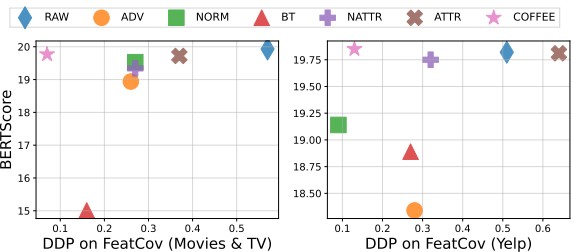

Figure 4: Trade-off between fairness and utility on Amazon Movies & TV (left) and Yelp (right) datasets. A method should give low DDP and high BERTScore (top-left corner of each plot) for better trade-offs.

(Lin, 2004) scores, we also employ the increasingly popular BERTScore (Zhang et al., 2020), known for better semantic alignment between target and generated text and stronger correlation with human judgments. We compute BERTScore using RoBERTa-large with re-scaled scores, and report F-1 scores for ROUGE and BERTScore. All utility metrics are *higher the better*. Lastly, we incorporate FeatCov into the utility metrics to evaluate the impact of various fairness attainment methods.

## 6.2 Comparison to Baselines

An ideal method should enhance fairness without hurting explanation utility. Due to space limits, we display complete results on Amazon Games in Table 2, and sample results for Amazon Movies and Yelp in Figure 4; full results are in Appendix E. As indicated in Table 2, COFFEE excels in fairness improvement across all metrics, significantly outperforming baselines. Moreover, comparing COFFEE and ATTR in terms of Ind-CF and Grp-CF validates the effectiveness of our fair policy learning scheme in optimizing CF constraints. Additionally, introducing $Q_{numtoken}$ and $Q_{feat}$ in COFFEE-NT bolsters fairness optimization in FeatCov, suggesting that enhanced quality measures can aid fairness optimization, echoing findings in (Bose and Hamilton, 2019). Crucially, COFFEE secures strong fairness outcomes while preserving high utility in explanation generation.

We plot results on Amazon Movies and Yelp in Figure 4, where we use DDP on FeatCov for fairness and BERTScore for utility. Notably, COFFEE achieves the best trade-off by drastically reducing DDP with minimal impact on BERTScore. ADV and NORM moderately improve fairness, albeit less significantly than COFFEE or at the cost of a greater decrease on utility. Lastly, BT also effectively enhances fairness, showing its interesting

Table 2: Comparison between COFFEE and baselines. BL stand for BLEU and RG denotes ROUGE. BLEU, ROUGE and BERTScore are in percentage values and others are in absolute values. The best results are boldfaced, and the second best are underlined. * indicates $p < 0.05$ for significance test over the second best baseline.

| PETER | Fairness on FeatCov | | | Utility | | | | | | |
|---|---|---|---|---|---|---|---|---|---|---|
| | Ind-CF↓ | Grp-CF↓ | DDP↓ | BL1↑ | BL4↑ | RG1↑ | RG2↑ | RGL↑ | BERT↑ | FeatCov |
| Amazon Games (User's Gender as Protected Attribute) | | | | | | | | | | |
| RAW | - | - | 1.18 | 8.82 | 1.49 | **19.95*** | **5.37*** | **15.54*** | **14.14** | 6.53 |
| ADV | - | - | 0.62 | 8.78 | 1.50 | 16.13 | 3.96 | 13.06 | 12.84 | 6.12 |
| NORM-F | - | - | 1.13 | 8.83 | 1.51 | 18.20 | 4.66 | 14.42 | 13.91 | 4.28 |
| BT | - | - | 0.65 | **10.43*** | 1.55 | 13.42 | 2.08 | 10.44 | 9.55 | 5.42 |
| NATTR | - | - | 0.63 | 8.76 | 1.52 | 16.55 | 3.86 | 13.02 | 12.72 | 5.26 |
| ATTR | 2.68 | 0.75 | 1.40 | 8.91 | 1.57 | 17.73 | 4.45 | 14.13 | 14.08 | 5.78 |
| COFFEE | 1.26 | 0.32 | 0.68 | 9.77 | **1.58** | 16.73 | 3.98 | 13.43 | 14.12 | 6.34 |
| COFFEE-NT | **1.15** | **0.01*** | **0.08*** | 10.23 | 1.24 | 17.38 | 3.70 | 13.62 | 14.02 | 6.27 |

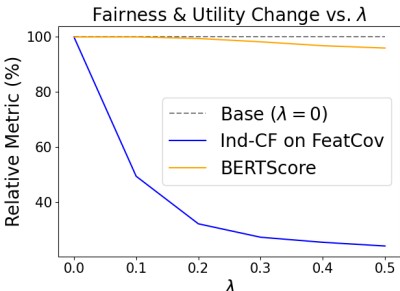

Figure 5: Relative Ind-CF and BERTScore ratio w.r.t. to $\lambda = 0$ in COFFEE. The Ind-CF and BERTScore of ATTR-PETER are 12.42 and 14.08, respectively.

normalization effect. But BT causes a dramatic sacrifice on utility, as the translation may eliminate much information about personal preference.

## 6.3 Effect of Tuning $\lambda$ in COFFEE

We further study the fairness-utility trade-off in COFFEE by tuning the weight $\lambda$ on fairness constraint in Eq. (4). We use Ind-CF on FeatCov for fairness evaluation which corresponds to the constraint that COFFEE directly optimizes, and use BERTScore for utility evaluation. We plot the results on Amazon Movies in Fig. 5, where the y-axis is the metric values ratio to the base results when $\lambda = 0$ (equivalent to ATTR). As $\lambda$ increases from 0 to 0.5, COFFEE significantly improves fairness, with a drop of about 77% in Ind-CF, while barely hurts the BERTScore, with a drop of about 3%. This study shows COFFEE's outstanding efficiency and effectiveness in fairness optimization.

## 7 Human Evaluation

We conduct human evaluations to justify the use of FeatCov for fairness and confirm COFFEE's efficacy. Human evaluators, recruited on Mechanical Turk, are asked to assess explanations generated from Amazon Movies' test data, as movies are

Table 3: Fairness and quality based on human evaluations of explanations. Smaller Ind-CF means better fairness. * indicates $p$-value <0.05 for paired t-test.

| | Ind-CF↓ | | Average↑ | |
|---|---|---|---|---|
| | RAW | COFFEE | RAW | COFFEE |
| Infor | 0.8476 | **0.5673*** | 2.6381 | **2.6712** |
| Detail | 0.8190 | **0.6442*** | **2.5238** | 2.3788 |
| Helpful | 0.8857 | **0.5786*** | **2.5904** | 2.5577 |

more universally understood. In each questionnaire, we present the title of a recommended movie for a user, randomly sampled from the dataset, along with two explanations: one generated for the real user and the other for a counterfactual user with a modified gender. Evaluators then assess each explanation from three criteria: *informativeness*, *detailedness*, and *helpfulness*. Each criterion is rated on a scale of 5: 1: "not at all", 2: "somewhat", 3: "moderate", 4: "very", 5: "extremely". We also collect evaluators' gender (only for aggregated analysis) to examine if humans with different genders evaluate the explanations differently. Rigorous privacy protection and quality control mechanisms are implemented. We compare the explanations generated from the raw model and COFFEE, with 150 valid questionnaires collected for each.

We measure fairness in the three human evaluated criteria of explanations. The results are shown in Table 3, where we use Ind-CF as the fairness metric. COFFEE can effectively achieve better fairness on human evaluated quality measures by improving fairness on FeatCov. Moreover, as indicated by the average quality measure values, the quality of explanations generated by COFFEE is comparable to that of the RAW model. These results demonstrate COFFEE's ability and practical use in achieving fairness in explanation generation.

# 8 Conclusion

We investigate the fairness problem in PTG by focusing on the bias in FeatCov of explanation generation. We propose COFFEE for achieving counterfactual fairness in explanation generation. Comprehensive experiments and human evaluations show the efficacy of COFFEE in achieving fairness without hurting the utility of explanations.

This work opens the new direction of quality fairness in PTG. We anticipate that this work will encourage researchers to investigate novel fairness notions in this problem, based on different quality measures, and conduct in-depth analyses on their social impacts. A promising direction is to involve humans in the loop for direct fairness optimization on human evaluated measures. We also plan to generalize the COFFEE to other PTG settings, such as conversational systems.

## Limitations

In this study, we formulated a counterfactual fairness constraint on explanation qualities, which is grounded in sampled explanations from both real and counterfactual worlds. We employed straightforward first moment matching to minimize the disparity between these two sample sets. Future work could explore the use of higher moment matching to more effectively align the quality distributions from the two worlds. For our experiments, we utilized FeatCov of explanations as a quality measure, given its simplicity and significance in the context of explainable recommendations. However, exploring other domain-specific measures (e.g., sentiment, emotion) to define $Q$ and assess COFFEE's versatility in different PTG settings would be intriguing. We have also achieved encouraging results in terms of sentiment fairness using COFFEE. Nevertheless, a more detailed discussion on this falls outside the scope of this paper's focus on quality fairness, and thus we earmark it for future investigation.

## Acknowledgements

This work was partially supported by NSF IIS-2007492 and NSF IIS-1838615.

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

## A  Applying COFFEE to NRT

There are mainly two modules in NRT (Li et al., 2017)—a MLP for rating prediction, and an RNN for explanation generation. Both modules share the same user and item embeddings mapped from input user and item IDs for personalization. In particular, the user and item embeddings are used in the initial hidden state of the RNN for explanation generation (see Figure 2 in (Li et al., 2017)). To apply COFFEE to NRT, we simply need to use the concatenation of disentangled user preference embedding and the attribute embedding instead of the holistic user embedding, exactly as introduced in Section 4.1. Similar to PETER, there are also three loss terms in the original NRT, corresponding to the explanation, context, and rating prediction (see Section 3 in (Li et al., 2017)). We denote the total loss of NRT by $\mathcal{L}_{nrt}$, which is used to replace $\mathcal{L}_{gen}$ in Eq. (3) for applying COFFEE.

## B  Experiment Setup

### B.1  Dataset Processing

The first two datasets are the *Video Games* and the *Movies & TV* categories from Amazon reviews. These two datasets only provide the names of users. We use a gender classification tool[6] to predict the gender of users from their names. To guarantee the accuracy of the attribute values, we only reserve the users whose names can be confidently classified as *male* or *female* and remove those classified as *unknown*. In the Yelp dataset, we use a restaurant's price range as the protected attribute. There are originally four price ranges ($, $$, $$$, $$$$) provided. However, we found that there are much less four-dollar restaurants than in the other price ranges, and thus we merge the four-dollar restaurants into the group of three-dollar ones for experiments. Using restaurant price as a sensitive attribute is based on the counterfactual argument: if a restaurant chose to raise/lower the price of its dishes without other changes, the quality of generated explanations shouldn't change. The rationale is that users may care more about experiences in expensive restaurants since they are paying more, and thus writing more detailed reviews. As shown in Table 4, the explanations for expensive restaurants have higher FeatCov and thus are more informative and detailed, which helps recommendations on them be accepted more often. However, this

---

[6]https://pypi.org/project/gender-guesser/

---

Table 4: Average FeatCov of explanations generated for restaurants of different price tags.

| FeatCov | $ | $$ | $$$ |
|---|---|---|---|
| Ground-Truth | 3.15 | 3.72 | 4.38 |
| RAW PETER | 3.26 | 3.68 | 4.03 |

will also leave the lower-priced restaurants at a disadvantage.

We split each dataset into training (80%), validation (10%), and testing (10%) sets, and ensure that there is at least one record in each subset for every user and item.

### B.2  Details of Baselines

• ADV (in-processing): Adversarially removing the sensitive information in user or item representations by adding a discriminator (Li et al., 2021b). We add discriminators on the user's (or item's) embedding in PETER and NRT to remove the information about protected attributes.

• NORM (pre-processing): We normalize the training data to remove the bias on group-level. With two demographic groups, we remove the explanations with the higher (lower) quality in the group with more reviews, until the difference of average quality between the two groups is below 10% of the original difference. With more than two groups, we recursively apply the procedure to the two groups with the maximum difference until the maximum difference is below the threshold. NORM removes less than 2% of training data on Amazon datasets, and less than 18% on Yelp.

• We pre-process the training data by translating the explanations to Chinese and then back to English. We use the multilingual model mBART (Tang et al., 2020) from EasyNMT[7] to perform the translation.

• ATTR: In order to evaluate the effectiveness of optimizing the fairness constraint in COFFEE, we use the model trained without the constraint ($\lambda = 0$ in Eq. (4)) as a baseline. We call the model ATTR, which means the attribute is disentangled from the user's or item's representations as introduced in Section 5 but without adding the fairness constraint.

• NATTR (post-processing): We disable the protected attribute token in ATTR for generating explanations during inference. Specifically, for NATTR-NRT, we replace the learned attribute embeddings

---

[7]https://github.com/UKPLab/EasyNMT

with random embeddings with values uniformly sampled in $[-1, 1]$ for explanation generation. For NATTR-PETER, we disable the other tokens' attention on the attribute token during inference.

## B.3 Evaluation Metrics

We evaluate both individual-level and group-level fairness, with and without counterfactual perspectives. The first metric denoted as Ind-CF evaluates the originally defined CF on individuals (Kusner et al., 2017), which is the same as the regularization term in Eq. (4) averaged over all user-item pairs on the test set $\mathcal{D}$:

$$\text{Ind-CF} \tag{8}$$
$$= \frac{1}{|\mathcal{D}|} \sum_{u,i \in \mathcal{D}} \big| \mathbb{E}[Q(y_{A \leftarrow a})|u,i] - \mathbb{E}[Q(y_{A \leftarrow a'})|u,i] \big|.$$

We also evaluate the counterfactual effect on group-level (Coston et al., 2020). For each attribute value $a$ and the corresponding demographic group, we counterfactually change the attribute value of all users in this group by $a' \neq a$, and calculate the change on the average quality of this group. We call this metric Group-CF:

$$\text{Group-CF} \tag{9}$$
$$= \frac{1}{|\mathcal{A}|(|\mathcal{A}|-1)} \sum_{a \in \mathcal{A}} \sum_{a' \neq a} \frac{1}{|\mathcal{D}_a|}$$
$$\left| \sum_{u,i \in \mathcal{D}_a} \mathbb{E}[Q(y_{A \leftarrow a})|u,i] - \sum_{u,i \in \mathcal{D}_a} \mathbb{E}[Q(y_{A \leftarrow a'})|u,i] \right|,$$

where $\mathcal{D}_a$ denotes the demographic group with attribute value $a$.

Finally, besides counterfactual metrics, we also evaluate COFFEE's generalization to improve group-wise fairness by the popular notion of demographic disparity (DDP) (Zafar et al., 2015):

$$\text{DDP} \tag{10}$$
$$= \frac{2}{|\mathcal{A}|(|\mathcal{A}|-1)} \sum_{a,a' \in \mathcal{A}}$$
$$\left| \frac{1}{|\mathcal{D}_a|} \sum_{u,i \in \mathcal{D}_a} \mathbb{E}[Q(y)|u,i] - \frac{1}{|\mathcal{D}_{a'}|} \sum_{u,i \in \mathcal{D}_{a'}} \mathbb{E}[Q(y)|u,i] \right|$$

All fairness metrics are *lower the better*, and the quality expectation on each user-item pair is calculated over $N = 3$ sampled explanations.

| COFFEE | PETER | | | NRT | | |
|---|---|---|---|---|---|---|
| | Games | Movies | Yelp | Games | Movies | Yelp |
| $\lambda$ | 0.2 | 0.2 | 0.2 | 0.3 | 0.1 | 0.1 |
| $\eta$ | 0.6 | 0.6 | 0.5 | 0.5 | 0.5 | 0.5 |

Table 5: Weight $\lambda$ for the fairness constraint and the promotion weight $\eta$ in reward calibration of COFFEE when applied to PETER and NRT on the three datasets.

## C  Experiment Settings

Here we elaborate the experiemntal protocols, model specifications, and hyper-parameters. For all models, we set the size of vocabulary to 20,000 by keeping the most frequent words. By default, we use top-5 sampling (Fan et al., 2018) as the decoding strategy, and the maximum decoding sequence length is 128.

For the raw PETER model (Li et al., 2021a), we mostly follow the original paper's hyper-parameters. The token embedding dimension is 512 and the dimension of feed-forward network is 2,048. The number of transformer layers and attention heads are both two. The dropout rate during training is 0.2. For NRT (Li et al., 2017), we follow the original paper and set the embedding dimension to 300 for all users, items and words. The dimension of hidden layers is 400. The dropout rate during training is 0.1. COFFEE consists of pre-training the ATTR models with disentangled attribute representations and then fine-tune them with the counterfactual fairness constraints. For ATTR-PETER, we just add an additional attribute token to PETER, with the same embedding dimension for the attribute. For ATTR-NRT, we disentangle the attribute representation by concatenating to the user or item embedding an attribute embedding of dimension 100. The other model specifications for ATTR-PETER and ATTR-NRT are the same as raw PETER and NRT. When applying the discriminator for removing the information about the protected attribute from user or item embeddings, we use a 2-layer MLP with hidden size of 512 as the attribute discriminator. For both ATTR-PETER and ATTR-NRT, the weight $\lambda_D$ of the adversarial loss is set to 0.5 on Amazon Games and Yelp datasets, and 0 on Amazon Movies. We iterate between one epoch of model training and one epoch of discriminator training until convergence. After the pre-training of ATTR models, we fix the user, item and attribute embeddings, remove the loss on rating prediction but keep the loss on explanation generation, and tune the model with the fairness constraint. The

Table 6: Kruskal-Wallis H test between FeatCov and quality and utility evaluated by human.

| K-W H test | | Infor | Detail | Help |
|---|---|---|---|---|
| All | *H-stats* | 39.44 | 45.67 | 23.83 |
| | *p-value* | $5.6e^{-8}$ | $2.9e^{-9}$ | $8.6e^{-5}$ |
| Male | H-stats | 24.28 | 28.07 | 14.37 |
| | *p-value* | $7.0e^{-5}$ | $1.2e^{-5}$ | $6.2e^{-3}$ |
| Female | H-stats | 16.58 | 15.74 | 12.24 |
| | *p-value* | $2.3e^{-4}$ | $3.4e^{-4}$ | $5.6e^{-3}$ |

weight $\lambda$ for the fairness constraint and the promotion weight $\eta$ in reward calibration are tuned for different models and datasets, and are listed in Table 5.

For model training, we use Adam as optimizer and the initial learning rate for training raw models and ATTR models is 1e-4. The initial learning rate is 1e-5 for COFFEE during fine tuning by the fairness constraint. By default, the batch size is 16. But we use batch size of 8 during the tuning phase of COFFEE when applying to PETER models, mainly due to memory issues. For training raw models and ATTR models, we evaluate the total loss on the validation set after each epoch, and use the epoch checkpoint when the total loss is higher in the next 5 consecutive epochs. When COFFEE fine-tunes the pre-trained ATTR models, we always use the checkpoint after a single epoch, which already yields promising results from COFFEE.

## D    Results from Kruskal-Wallis H Tests

We perform Kruskal-Wallis H test between Feat-Cov and three human evaluated measures including *Informativeness*, *Detailness*, and *Helpfulness*. The results are shown in Table 6, where a H-stats larger than one means a strong correlation, and a p-value smaller than 0.05 means the correlation is significant. FeatCov strongly and significantly correlate with humans' (both male and female) judgement of the quality and helpfulness of explanations. Thus, the bias can lead to unfair treatments to users of different genders, as the quality of explanations, evaluated by FeatCov in this case, may affect the utility of explanations to users.

## E    Additional Results

We show the complete results on the Amazon Movies and Yelp datasets based on the PETER model in Table 7. Importantly, COFFEE achieves strong fairness results while maintaining high utility on explanation generation. Although COFFEE may slightly drop the utility compared to the baselines, it can sometimes even outperform the baselines, as shown in the results on Amazon Movies and Yelp datasets. This is because the disentanglement mechanism enables better representation learning (Ma et al., 2019; Zheng et al., 2021) and increases the flexibility and accuracy of interactions between the user and item for better explanation generation.

We also present some examples of generated explanations on Amazon Games in Table 8. In particular, we select a male user *Chadwick* who wrote detailed and informative reviews with high Feat-Cov and a female user *Noemi* who wrote generic reviews with low FeatCov for the same item "*Wii Nunchuk Controller*". The raw PETER model follows the ground-truth reviews and generate detailed explanation for *Chadwick* but short and generic explanation for *Noemi*. In contrast, after applying COFFEE to improve fairness, the model tend to generate and more descriptive and informative explanation for *Noemi*. These examples demonstrate the effectiveness of COFFEE in generating fair and high-quality explanations for users with different protected attribute values.

## E.1    Entanglement of Protected Attribute with Preference

In Section 4.1, we employ a disentanglement approach where each user's representation is separated into two components: the attribute representation and the preference representation. This setup enables us to perform counterfactual inference by switching the attribute representation from one attribute value to another. Following this method, we analyzed how much changing a user's learnt attribute representation will influence the FeatCov of generated explanations. Take the Amazon Games dataset for example, without imposing any fairness constraints, we observed that the average FeatCov change is 2.68 when altering the attribute representation. Furthermore, over $92\%$ of users exhibit a FeatCov change greater than 1. This indicates that, in most cases, the attribute and preference representations are intertwined, and the disentanglement approach effectively separates these two aspects.

Table 7: Comparison between COFFEE and baselines based on the PETER model. BL stand for BLEU and RG denotes ROUGE. BLEU, ROUGE and BERTScore are in percentage values and others are in absolute values. The best results are boldfaced, and the second best are underlined. * indicates $p < 0.05$ for significance test over the second best baseline.

| PETER | Fairness on FeatCov | | | Utility | | | | | | | |
|---|---|---|---|---|---|---|---|---|---|---|---|
| | Ind-CF↓ | Grp-CF↓ | DDP↓ | BL1↑ | BL4↑ | RG1↑ | RG2↑ | RGL↑ | BERT↑ | FeatCov |
| Amazon Movies & TV (User's Gender as Protected Attribute) | | | | | | | | | | | |
| RAW | - | - | 0.57 | 11.25 | **2.20** | 18.04 | 4.53 | 15.48 | 19.93 | 5.61 |
| ADV | - | - | 0.26 | 11.08 | 2.16 | 17.20 | 4.32 | 14.37 | 18.94 | 5.14 |
| NORM | - | - | 0.27 | 11.23 | 2.17 | 17.46 | 4.42 | 15.00 | 19.67 | 4.34 |
| BT | - | - | 0.16 | 11.25 | 2.18 | 12.91 | 1.66 | 11.02 | 15.01 | 4.13 |
| NATTR | - | - | 0.27 | 11.24 | 2.19 | 17.35 | 4.40 | 14.92 | 19.34 | 4.71 |
| ATTR | 1.47 | 0.10 | 0.37 | 11.19 | 2.16 | 17.43 | **4.63** | **15.98** | 19.72 | 5.31 |
| COFFEE | **1.13** | 0.01 | 0.07 | **11.28** | 2.12 | 17.77 | 4.60 | 15.35 | **20.13*** | 5.45 |
| COFFEE-NT | 1.16 | 0.01 | **0.00*** | 10.05 | 1.39 | **19.34*** | 4.56 | 15.86 | 19.52 | 5.08 |
| Yelp (Restaurant's Price as Protected Attribute) | | | | | | | | | | | |
| RAW | - | - | 0.51 | 10.24 | 1.57 | 18.71 | 3.11 | 14.64 | 19.82 | 3.78 |
| ADV | - | - | 0.28 | 9.87 | 1.60 | 17.25 | 2.64 | 13.99 | 18.34 | 3.18 |
| NORM | - | - | 0.09 | 9.96 | 1.54 | 17.56 | 2.81 | 14.10 | 19.67 | 3.06 |
| BT | - | - | 0.27 | 10.69 | 1.61 | 16.42 | 2.36 | 13.29 | 18.89 | 3.20 |
| NATTR | - | - | 0.32 | 10.08 | 1.55 | 18.53 | 3.06 | 14.57 | 19.75 | 3.37 |
| ATTR | 1.50 | 0.44 | 0.64 | 10.00 | 1.53 | **18.83*** | **3.16** | **14.75** | 19.81 | 3.52 |
| COFFEE | 0.84 | 0.08 | 0.13 | 11.17 | **1.77** | 17.97 | 3.00 | 14.39 | **20.58*** | 3.68 |
| COFFEE-NT | **0.54*** | **0.03** | **0.03*** | **11.61*** | 1.72 | 17.41 | 2.93 | 14.31 | 20.21 | 3.61 |

Table 8: Generated explanations by different models on Amazon Games. We select one male user *Chadwick* and one female user *Noemi*, and present the explanations for them on the same item "*Wii Nunchuk Controller*".

| PETER | user: *Chadwick*, item: *Wii Nunchuk Controlle* | user: *Noemi*, item: *Wii Nunchuk Controller* |
|---|---|---|
| Ground-truth | this review is for the white wii nunchuk controller. it is a necessary component in most wii games, and attaches to the bottom port of every wii remote. it is well-designed, sturdy, and comfortable to hold. the price is relatively low for such a controller. | great controller, arrived as expected. |
| RAW | i bought it to play wii u and this is a great addition to the wii remote. the gamepad is a nice addition to the wii u gamepad. the wii u is a must have for any wii u console owner. | the controller worked very well. |
| COFFEE | i bought this controller to use the nunchuck. this is a great controller for wii - u owners especially the wii u. the controller is very responsive and the triggers are awesome. | i bought this controller for my wii u. it was exactly what it would be expected. i love the feel of the box and it works great. |

## E.2 Results based on NRT

We present the complete results based on the NRT model in Table 9. Similar to the results on PETER, COFFEE when applied to NRT can significantly outperform the baselines in terms of fairness improvement. Again, when optimizing both $Q_{numtoken}$ and $Q_{feat}$ together, COFFEE achieves the best fairness improvements, verifying the effect of correlations between different quality measures. In general, PETER can generate better explanations than NRT, e.g., on BERTScore, with or without fairness optimizations, showing the capability of transformers over RNNs. However, COFFEE still maintains high generation utility based on NRT when compared to baselines, showing its advantage in generalizing the results to different models.

The observations and conclusions are similar to the PETER based results. These results show COFFEE's ability to generalize the fairness improvement to different types of models, and indicate its flexibility and practicality in real-world uses.

Table 9: Comparison between COFFEE and baselines based on the NRT model. BL stand for BLEU and RG denotes ROUGE. BLEU, ROUGE and BERTScore are in percentage values and others are in absolute values. The best results are boldfaced, and the second best are underlined. * indicates $p < 0.05$ for significance test over the second best baseline.

| NRT | Fairness on $Q_{feat}$ | | | Utility | | | | | | |
|---|---|---|---|---|---|---|---|---|---|---|
| | Ind-CF↓ | Grp-CF↓ | DDP↓ | BL1↑ | BL4↑ | RG1↑ | RG2↑ | RGL↑ | BERT↑ | FeatCov |
| Amazon Games (User's Gender as Protected Attribute) | | | | | | | | | | |
| RAW | - | - | 1.42 | 8.33 | 1.30 | 19.21 | 3.97 | **14.35*** | 13.79 | 6.48 |
| ADV | - | - | 1.25 | 8.13 | 1.29 | 17.67 | 3.37 | 13.27 | 12.85 | 6.08 |
| NORM | - | - | 1.79 | 8.17 | 1.26 | **19.89*** | **4.27*** | 13.65 | 13.49 | 4.80 |
| BT | - | - | 1.47 | 8.34 | 1.17 | 16.44 | 2.71 | 12.23 | 10.73 | 5.29 |
| NATTR | - | - | 0.12 | 8.39 | **1.39** | 13.07 | 2.71 | 10.83 | 11.11 | 5.83 |
| ATTR | 6.70 | 1.46 | 1.65 | **8.98** | 1.29 | 18.94 | 3.89 | 14.29 | **13.84** | 6.23 |
| COFFEE | 1.81 | 0.52 | 0.51 | 8.91 | 1.30 | 17.94 | 3.50 | 13.63 | 13.83 | 6.30 |
| COFFEE-NT | **0.77*** | **0.02*** | **0.04** | 8.67 | 1.26 | 17.11 | 3.11 | 12.46 | 13.69 | 6.27 |
| Amazon Movies & TV (User's Gender as Protected Attribute) | | | | | | | | | | |
| RAW | - | - | 0.46 | 10.55 | 1.86 | 18.37 | 3.98 | 15.16 | 19.20 | 5.46 |
| ADV | - | - | 0.24 | 10.48 | 1.83 | 18.44 | 3.95 | 15.10 | 19.09 | 5.18 |
| NORM | - | - | 0.46 | 10.53 | 1.91 | 18.09 | 3.89 | 14.96 | 19.03 | 4.62 |
| BT | - | - | 0.43 | 10.63 | 1.62 | 14.84 | 1.87 | 11.95 | 14.36 | 5.06 |
| NATTR | - | - | 0.33 | 9.84 | 2.07 | 15.90 | 3.37 | 13.77 | 16.57 | 4.75 |
| ATTR | 1.74 | 0.14 | 0.48 | 10.53 | 1.84 | **18.68*** | **4.10** | **15.34*** | 19.21 | 5.07 |
| COFFEE | 1.23 | 0.04 | 0.24 | **11.05** | **2.08** | 17.90 | 4.02 | 15.11 | 19.27 | 5.48 |
| COFFEE-NT | **1.10*** | 0.04 | **0.19*** | 10.52 | 2.02 | 17.04 | 3.92 | 14.77 | **19.35*** | 5.27 |
| Yelp (Restaurant's Price as Protected Attribute) | | | | | | | | | | |
| RAW | - | - | 0.56 | 10.09 | 1.49 | 19.14 | **3.17** | 14.83 | 19.50 | 3.76 |
| ADV | - | - | 0.21 | 10.35 | 1.97 | 16.78 | 2.49 | 13.14 | 17.88 | 3.24 |
| NORM | - | - | 0.15 | 9.88 | 1.42 | **19.20** | 3.14 | **14.85** | 19.48 | 3.04 |
| BT | - | - | 0.30 | **10.56*** | 1.52 | 16.44 | 2.31 | 13.24 | 18.95 | 3.10 |
| NATTR | - | - | 0.19 | 10.38 | **2.23*** | 15.89 | 2.26 | 12.58 | 17.26 | 3.04 |
| ATTR | 1.58 | 0.42 | 0.50 | 10.08 | 1.85 | 18.92 | 3.09 | 14.72 | **19.74** | 3.22 |
| COFFEE | 0.93 | 0.17 | 0.10 | 10.25 | 1.75 | 18.34 | 2.92 | 14.33 | 19.49 | 3.70 |
| COFFEE-NT | **0.69*** | **0.09*** | **0.02** | 10.45 | 2.07 | 18.65 | 2.86 | 14.25 | 19.67 | 3.52 |