# OpenReview forum: "COFFEE: Counterfactual Fairness for Personalized Text Generation in Explainable Recommendation"
_EMNLP/2023/Conference — EMNLP 2023 Main_

### Official Review · Reviewer_iBGj · 2023-08-04

**Soundness:** 3

**Excitement:**

4: Strong: This paper deepens the understanding of some phenomenon or lowers the barriers to an existing research direction.

**Paper Topic And Main Contributions:**

This paper investigates the fairness of Personalized Text Generation (PTG) in the context of personalized explanation generation for recommendations. The authors identify that bias inherent in user-written text, often used for PTG model training, can inadvertently associate different levels of linguistic quality with users' protected attributes, leading to unfair treatment. To address this, they introduce a general framework to achieve measure-specific counterfactual fairness in explanation generation. The effectiveness of their method is demonstrated through extensive experiments and human evaluations.

**Reasons To Accept:**

This paper provides a very interesting and tractable structural model of counterfactual fairness in recommendations. Technically, the use of disentangled representations and policy learning are nice contributions.

**Reasons To Reject:**

I really like the theoretical part of this paper, but the application is not great.

This paper shows that men and women are different in how they interact with Amazon reviews, which results in differences in the features of  recommendations presented. This is immediately interpreted as harmful to women users, and so an extremely invasive system is proposed to try to give men and women identical recommendations.

Is FeatCov really quality? That is not justified nearly enough. Do men like a higher FeatCov than women? That possibility is not even entertained.

Finally, counterfactual fairness relies on strong structural assumptions. Those are not justified well in this paper.

**Reproducibility:**

4: Could mostly reproduce the results, but there may be some variation because of sample variance or minor variations in their interpretation of the protocol or method.

**Reviewer Confidence:**

3: Pretty sure, but there's a chance I missed something. Although I have a good feel for this area in general, I did not carefully check the paper's details, e.g., the math, experimental design, or novelty.

**Typos Grammar Style And Presentation Improvements:**

Figure 1 needs to fix the Y axis scale so you can see the difference between the left and right plot.

Not every paper needs a cute acronym like COFFEE.

---

> ### Author Rebuttal · Authors · 2023-08-29
>
> Dear reviewer iBGj,
>
>
> Thank you very much for your comments and suggestions for our paper! We believe that most of your concerns/questions can be addressed properly based on our user study presented in the paper. Below please find detailed responses to your concerns. We sincerely hope our responses can address your concerns.
>
>
> **Regarding the study of different interaction patterns with Amazon reviews**
>
>
> Thank you for the insightful comment. We would like to provide some justification here.
>
> Our claims and motivations are supported by both empirical observations and comprehensive user studies. We first observed the different behavior patterns of Amazon reviews from male and female users, which are inherited by the model generated explanations that have higher FeatCov for male users than female users. To confirm whether such bias in explanations is harmful, we performed rigorous user studies (firstly introduced in Figure 2 and detailed in Section 7 and Appendix D) to examine how real humans evaluate the usefulness of explanations. The human evaluation results suggest that humans’ judgments of explanation quality are highly correlated with FeatCov (Figure 2), and the bias is also observed in human evaluation measures including informativeness, detailedness, helpfulness (Table 3). Thus we confirmed the harmfulness of such bias which motivated our research for fairness-aware explanation generation. The proposed COFFEE generally addresses a class of such fairness issues in the broad personalized text generation setting. As suggested in Section 6, we experimented with different datasets and models and showed that COFFEE is able to address fairness issues with respect to different sensitive attributes (besides gender).
>
>
> **Regarding using FeatCov as a quality measure**
>
>
> We actually experimented with different automatic evaluation metrics for evaluating explanations, including FeatCov, length, grammar, redundancy, structure, etc. Then we correlated these metrics with human evaluations of explanation quality (informativeness, detailedness, helpfulness), and found FeatCov aligns well and highly correlated with human evaluations. Therefore, we chose FeatCov as an automatic quality evaluation measure for fairness improvement. Thank you for pointing it out. We will provide further discussion in our revision.
>
>
> **Regarding structural assumptions for counterfactual fairness**
>
>
> We agree that achieving counterfactual fairness requires casual structure assumptions. In fact, in practical applications of counterfactual fairness (CF), we usually approximate CF by imposing a fairness constraint in the optimization objective (see Section 3.2, 3.3 in [1]). We followed this recipe in developing CF for explanation generation as in Section 3 and Section 4.2 in our paper. Thank you for the suggestion, and we will make this clearer in the revision of the paper.
>
>
> [1] Russell, Chris, et al. "When worlds collide: integrating different counterfactual assumptions in fairness." Advances in neural information processing systems 30 (2017).
>
>
> **Regarding other Presentation Improvements**
>
> Thank you for the great suggestions. We will revise accordingly.

---

### Official Review · Reviewer_yP6b · 2023-08-04

**Typos Grammar Style And Presentation Improvements:** N/A
**Soundness:** 2

**Excitement:**

3: Ambivalent: It has merits (e.g., it reports state-of-the-art results, the idea is nice), but there are key weaknesses (e.g., it describes incremental work), and it can significantly benefit from another round of revision. However, I won't object to accepting it if my co-reviewers champion it.

**Missing References:**

N/A

**Paper Topic And Main Contributions:**

This paper addressed fairness in personalized explanation generation for explainable recommendations. The authors conducted an analysis of unfairness in personalized explanation generation and proposed a COFFEE model, which took a causal perspective to enforce fairness. The COFFEE model consisted of a counterfactual inference module, a disentangled learning module, and a policy learning module. Extensive experiments showed the effectiveness of the COFFEE model.

While the topic of counterfactual fairness in explainable recommendation is worth studying, I have some concerns with this paper. Specifically, I have identified some incorrect statement in this manuscript, which make me unable to recommend it for acceptance.


**Questions For The Authors:**

N/A

**Reasons To Accept:**

1. The topic of counterfactual fairness in explainable recommendations is worth studying.
2. Extensive experiments prove the effectiveness of the COFFEE model.
3. Based on my experience, the disentangled learning module in the proposed model appears to be technically sound. I have confidence in the validity of the experimental results.


**Reasons To Reject:**

1.	My biggest concern is the Counterfactual Inference (CI) module. As far as I know, all fairness-aware models are designed to remove sensitive attribute information, but here the proposed model takes a joint input of sensitive attributes and user-product IDs, which seems counterintuitive.
Additionally, I'm puzzled by the fact that existing research has shown that even models that do not input sensitive attributes can still contain sensitive information in the user representations. From a causal perspective, this would mean that sensitive attributes (such as gender) have a causal effect on user representation, and user representation has a causal effect on the output. Therefore, the explicit input of sensitive attributes in this paper should be understood as 'sensitive attributes have a direct causal effect on user representation, and both user representation and sensitive attributes have a direct causal effect on the output'. I believe that the edge 'sensitive attributes have a direct causal effect on user representation' objectively exists. Therefore, if we want to perform CI, simply modifying the input sensitive attribute value without considering changes on user representation would not truly constitute CI.
2.	Given my doubts about the validity of counterfactual inference, I am also skeptical of the metrics used in the experimental section that involve CI.
3.	I also have concerns about the introduction section. The authors claim to have solved a problem in the field of personalized text generation, but in reality, the proposed model seems to be specific to the sub-field of generating natural language explanations for recommendations. It's important for the authors to explain the difference between these two areas or clarify why they chose to write it this way.
4.	I have some concerns regarding the experimental setup, particularly regarding why the price range ($, $$, $$$) of a restaurant is considered protected information in the Yelp dataset. In my experience, price is an essential factor to consider when evaluating a restaurant as it can determine its affordability. It would be helpful to have more information on why the authors consider the price range as protected information.


**Reproducibility:**

3: Could reproduce the results with some difficulty. The settings of parameters are underspecified or subjectively determined; the training/evaluation data are not widely available.

**Reviewer Confidence:**

4: Quite sure. I tried to check the important points carefully. It's unlikely, though conceivable, that I missed something that should affect my ratings.

---

> ### Author Rebuttal · Authors · 2023-08-29
>
> Dear reviewer yP6B,
>
> Thank you very much for your efforts in reviewing our paper and sharing your comments and concerns! We believe that there are significant mis-understandings of the fairness literature and our method. Thus we kindly request that the reviewer can carefully check our rebuttal and the corresponding sections in the submission. Below please find detailed responses to your concerns. We sincerely hope you can increase the score if the concerns are addressed. Thanks!
>
> **Regarding the counterfactual inference (CI) module**
>
> We would like to provide clarifications regarding some potential misunderstandings of the fairness notions and our proposed solution.
>
> First of all, not all fairness-aware methods are designed to remove sensitive attributes [7,8]. Most fairness works aim to improve specific fairness notions, such as group-wise demographic parity, individual fairness, and counterfactual fairness [2-7,12]. There are mainly three categories of fairness methods in literature, pre-, in-, and post-processing methods [4,7,8]. Neither pre- nor post processing methods require removing sensitive attributes. Only a small portion of in-processing methods propose to remove sensitive attribute information to learn attribute-independence representations [11,13], which, however, does not have guarantees in optimizing a specific fairness notion. In fact, the most popular in-processing method to achieve fairness is to impose a constraint or regularization corresponding to a specific fairness notion during optimization [1,2,3,10,12], which is also what our method COFFEE is designed for.
>
> In the literature of Counterfactual Fairness (CF), removing sensitive information has been explored in some papers to achieve CF, which however does not follow the exact CF definition and thus does not have any guarantee in optimizing CF [5,10].
>
> In lines 262-272, we discussed the methods that simply remove sensitive attribute information from representations, and why we are taking a more controlled way to improve fairness. In particular, although directly removing sensitive information from representations can potentially improve fairness in general, it also blindly removes all desired or undesired characteristics from the sensitive attribute and detrimentally affects the utility and performance of these representations for downstream tasks. For example, when explaining why a user might like a movie, a male user and a female user may prefer different aspects/factors (related to their gender) of the movie being highlighted in the explanation. Thus the personalized explanations should be related to their gender, but they should not receive explanations of different quality (informativeness, detailedness, helpfulness, etc.) because of their gender.
>
> Our goal here is to make sure that the explanations are still personalized to their gender, but the *quality* of the explanations should not depend on their gender. Therefore, we are taking a more controlled approach in improving fairness through CI with respect to any specific quality measures of generated explanations (as formulated in Section 3), while maintaining the performance of personalized explanation generation. The advantage of our approach is also shown in the experiment results. In Table 2, compared to ADV which simply removes sensitive attribute information from representations, COFFEE achieves superior performance in improving fairness while maintaining high utility of the explanations.
>
> The sensitive attribute is not exactly a direct input to the model for explanation generation, but is essentially a value token for manipulating the representations which are used for the generation. We use this sensitive attribute token to control how the representation is changed to reflect the attribute value change for CI, which is a commonly used method in controlled text generation (see lines 274-291).
> More specifically, the manipulation of the sensitive attribute value is required for performing CI to study how the model output changes according to the attribute value change [1,6,10,11]. In our method, we need to manipulate the sensitive attribute value on the learned representations. To achieve this, in Section 4.1 (lines 293-306), we propose to first learn disentangled representations for each user $r_u^a = [r_a, r_u]$, which consist of two parts: $r_a$ for the attribute representation of value $a$, and $r_u$ for the preference representation that is orthogonal to $r_a$. When performing CI of $A \leftarrow a’$ on user $u$ with true attribute value $a$, we swap this user’s attribute representation from $r_a$ to $r_a’$ to form $r_u^{A \leftarrow a’} = [r_a’, r_u]$. This modified representation $r_u^{A \leftarrow a’}$ is used to generate the counterfactual explanation and construct the CF constraint. Note that the input of the input sensitive attribute token is mainly used to indicate the CI performed on the user’s representation, which is used for explanation generation.
>
> [1] Huang, Po-Sen, et al. "Reducing Sentiment Bias in Language Models via Counterfactual Evaluation." Findings of the Association for Computational Linguistics: EMNLP 2020. 2020. \
> [2] Zemel, Rich, et al. "Learning fair representations." International conference on machine learning. PMLR, 2013.\
> [3] Jiang, Ray, et al. "Wasserstein fair classification." Uncertainty in artificial intelligence. PMLR, 2020.\
> [4] Du, Mengnan, et al. "Fairness in deep learning: A computational perspective." IEEE Intelligent Systems 36.4 (2020): 25-34.\
> [5] Kusner, Matt J., et al. "Counterfactual fairness." Advances in neural information processing systems 30 (2017).\
> [6] Hardt, Moritz, Eric Price, and Nati Srebro. "Equality of opportunity in supervised learning." Advances in neural information processing systems 29 (2016).\
> [7] Mehrabi, Ninareh, et al. "A survey on bias and fairness in machine learning." ACM computing surveys (CSUR) 54.6 (2021): 1-35.\
> [8] Li, Yunqi, et al. "Fairness in Recommendation: Foundations, Methods and Applications." ACM Transactions on Intelligent Systems and Technology.\
> [9] Garg, Sahaj, et al. "Counterfactual fairness in text classification through robustness." Proceedings of the 2019 AAAI/ACM Conference on AI, Ethics, and Society. 2019.\
> [10] Russell, Chris, et al. "When worlds collide: integrating different counterfactual assumptions in fairness." Advances in neural information processing systems 30 (2017).\
> [11] Bose, Avishek, and William Hamilton. "Compositional fairness constraints for graph embeddings." International Conference on Machine Learning. PMLR, 2019.\
> [12] Madras, David, et al. "Learning adversarially fair and transferable representations." International Conference on Machine Learning. PMLR, 2018.\
> [13] Elazar, Yanai, and Yoav Goldberg. "Adversarial removal of demographic attributes from text data." arXiv preprint arXiv:1808.06640 (2018).
>
> **Regarding the generality claim in introduction section**
>
> Our proposed method COFFEE is general to address fairness issues in personalized text generation (PTG). Focusing on explanation generation does not impose application-driven assumptions or specific adaptations of our method specific to this setting. We focused on the experiments in explanation generation mainly because it is a typical and widely studied setting of PTG, and there exists various models and datasets to experiment with.
>
> COFFEE can be easily applied to other PTG settings as well. For example, in personalized post generation, we can also apply COFFEE by learning disentangled representations (Section 4.1) of users and impose the counterfactual constraint for post generation model training (Section 4.2).
>
> We thank the reviewer for the great suggestion. We will revise accordingly to further clarify this.
>
> **Regarding the experiment setup on Yelp dataset**
>
> We consider the price range in Yelp restaurants because we observed that *the quality of generated explanations* are generally higher for pricer restaurants than for cheaper restaurants. Again, please note that we are not trying to remove the price information in generated explanations, but to make sure the quality of explanations (informativeness, detailedness, helpfulness) are agnostic to the price of restaurants, which otherwise will unfavor restaurants with a lower price. Customers can still choose restaurants according to the price, but the quality of system-generated explanations should not affect their decisions. In other words, the price range of restaurants is a sensitive attribute for *the quality of explanations* generated for different restaurants.
>
>
> Thanks again for your time reviewing our work! Please let us know if you have any further questions, and we are happy to discuss further.

---

### Official Review · Reviewer_VQ9B · 2023-08-05

**Soundness:** 4

**Excitement:**

4: Strong: This paper deepens the understanding of some phenomenon or lowers the barriers to an existing research direction.

**Missing References:**

a. https://aclanthology.org/2022.acl-long.20/;

b. https://dl.acm.org/doi/abs/10.1145/3477495.3532039;

c. https://dl.acm.org/doi/abs/10.1145/3523227.3546767;

d. https://dl.acm.org/doi/abs/10.1145/3580488.

**Paper Topic And Main Contributions:**

This paper studies the quality fairness problem in explanation generation for recommendations. Due to the bias of users’ protected attributes (e.g. gender) that exists in the training data, the quality of model-generated explanations may vary accordingly as attributes change. A general framework named COFFEE is proposed to achieve quality-measure-specific counterfactual fairness. In COFFEE, the user’s protected attribute value is disentangled from the user’s representation via adversarial training, and is instead modeled with a separate control token; the explanation generator is then trained according to a counterfactual fairness constraint applied to model-generated explanations, so that the explanation quality is encouraged to change minimally when the attribute value is altered counterfactually; that constraint is expressed as rewards for RL and the model is optimized via policy gradient. Experiments show the effectiveness of COFFEE in achieving counterfactual fairness while maintaining the utility of explanations.

**Questions For The Authors:**

A.	Line 126: Technically, any numerical quality measure can be plugged into COFFEE; however, this paper lacks experiments with a second quality measure other than FeatCov, so it might be insufficient to claim the generality of COFFEE.

B.	Line 258: I wonder how much the user’s protected attribute is entangled with their preference and is implicitly encoded in the representations – are they significant enough? Possible experiments to do include measuring the percentage of entangling cases, and ablation studies on “Sec 4.1 Disentangled Attribute Representation”.

C.	Line 314: What if there are more than two attributes? Formula (3) seems to work for binary attributes only.

D.	Line 412: I’m not sure if PETER is still the SOTA model in 2023 – there are a few follow-up works (listed below) of PETER, which I guess are supposed to be superior to PETER. More experiments with those advanced models are needed to verify the effectiveness of COFFEE more completely. (a. https://aclanthology.org/2022.acl-long.20/; b. https://dl.acm.org/doi/abs/10.1145/3477495.3532039; c. https://dl.acm.org/doi/abs/10.1145/3523227.3546767; d. https://dl.acm.org/doi/abs/10.1145/3580488.)

E.	Line 460: Are there any experimental results that can support this claim?

F.	Line 515: Given that Figure 2 shows a strong positive correlation between FeatCov and human-evaluated quality measures, after adding $Q_{numOfToken}$, have authors verified the existence of that correlation – does it remain strong or become weak?

**Reasons To Accept:**

1.	This paper is the first attempt at the quality fairness problem in explanation generation for recommendations.

2.	The mathematics looks sound, and the experiments are extensive.

3.	This paper is well-organized, and the writing is easy to follow.

**Reasons To Reject:**

(Please check the questions for details)

1.	The generality of the proposed framework appears to be limited.

2.	The experiments lack ablation studies.

3.	Some parts of the experiments are incomplete.

4.	Some parts of the paper presentation can be improved.

**Reproducibility:**

3: Could reproduce the results with some difficulty. The settings of parameters are underspecified or subjectively determined; the training/evaluation data are not widely available.

**Reviewer Confidence:**

3: Pretty sure, but there's a chance I missed something. Although I have a good feel for this area in general, I did not carefully check the paper's details, e.g., the math, experimental design, or novelty.

**Typos Grammar Style And Presentation Improvements:**

I think this paper does not have much to do with causality, so it might be misleading to mention the causal perspective in Line 103. Similarly, as explanation generation for recommendations is only one of the various use cases of personalized text generation, overclaiming the research scope by mentioning PTG in the paper title and introduction might hurt the paper's readability.

---

> ### Author Rebuttal · Authors · 2023-08-29
>
> Dear reviewer VQ9B,
>
>
> Thank you very much for your time and effort in reviewing our paper! We believe most of the concerns are due to lack of clarity and missing details that we decided not to include due to the page limit. Your questions also help us further clarify things and improve the paper. Below please find our detailed responses to your questions and we will revise the paper accordingly. We sincerely hope our responses can address your concerns and help you consider increasing the score.
>
>
> **Regarding generality of COFFEE**
> >Technically, any numerical quality measure can be plugged into COFFEE; however, this paper lacks experiments with a second quality measure other than FeatCov, so it might be insufficient to claim the generality of COFFEE.
>
>
> First of all, our primary objectives for fairness optimization revolve around enhancing human-evaluated quality criteria such as *informativeness*, *detailedness*, and *helpfulness* of explanations, as extensively discussed in Section 7. We ultimately opted for FeatCov because it demonstrates a strong correlation with these human-evaluated quality criteria, as evidenced in Figure 2.
>
> Actually, we also conducted experiments using the length of explanations as another quality metric. Notably, in the Amazon datasets, we observed that generated explanations tend to be longer for male users compared to female users. Similarly, in the Yelp dataset, explanations are generally longer for higher-priced restaurants than their less pricey counterparts. To address fairness concerns related to explanation length, we incorporated Q_{numToken} into COFFEE for fairness optimization. The results obtained for the Amazon Games dataset are presented below.
>
> ### Fairness on Length of explanations
> |         | Ind-CF  | Grp-CF | DDP   | BERT  |
> |---------|---------|--------|-------|-------|
> | RAW     | -       | -      | 5.84  | 14.14 |
> | ADV     | -       | -      | 3.04  | 12.84 |
> | NORM    | -       | -      | 5.67  | 14.08 |
> | BT      | -       | -      | 4.00  | 9.55  |
> | NATTR   | -       | -      | 3.24  | 12.72 |
> | ATTR    | 12.42   | 3.63   | 6.81  | 14.08 |
> | COFFEE  | 3.87    | 0.47   | 1.09  | 14.09 |
>
> As illustrated, COFFEE successfully mitigated the bias in length of explanations while preserving the quality of the generated content, as assessed by BERTScore. Similar results are observed across the other two datasets. While we did not include these results initially, it's important to note that explanation length does not exhibit as strong a correlation with human evaluations as FeatCov. Nevertheless, to demonstrate the broad applicability and effectiveness of COFFEE, we will incorporate these results into the appendix as supplementary evidence of COFFEE's capabilities.
>
>
> **Regarding experiments on the entanglement of protected attribute with preference**
> >I wonder how much the user’s protected attribute is entangled with their preference and is implicitly encoded in the representations – are they significant enough? Possible experiments to do include measuring the percentage of entangling cases, and ablation studies on “Sec 4.1 Disentangled Attribute Representation”
>
>
> Thank you for the suggestion. In Section 4.1, we employ a disentanglement approach where each user's representation is separated into two components: the attribute representation and the preference representation. This setup enables us to perform counterfactual inference by switching the attribute representation from one attribute value to another.
>
>
> Following this method, we analyzed how much changing a user’s learnt attribute representation will influence the FeatCov of generated explanations. Take the Amazon Games dataset for example, without imposing any fairness constraints, we observed that the average FeatCov change is 2.68 when altering the attribute representation. Furthermore, over 92% of users exhibit a FeatCov change greater than 1. This indicates that, in most cases, the attribute and preference representations are intertwined, and the disentanglement approach effectively separates these two aspects. In our revision, we’ll provide more comprehensive and detailed ablation studies to further elucidate these findings.
>
>
> **Regarding the cases with non-binary attributes**
> >What if there are more than two attributes? Formula (3) seems to work for binary attributes only.
>
> Thank you for your question. We'd like to clarify that COFFEE is designed to accommodate non-binary attributes with any number of value choices, as explicitly mentioned in Section 4.1 and reiterated in footnote 3 after line 331. Specifically, the constraint outlined in Eq. (3) is articulated as $\mathbb E[Q(Y_{A\leftarrow a})] = E[Q(Y_{A\leftarrow a’})], \forall a’\in\mathcal A, a’ \neq a$. Here, it's important to note that the set of attribute values $\mathcal A$ is not limited to binary choices. We will further clarify this in the revision.
>
>
> **Regarding the use of PETER as SOTA for experiments**
> >I’m not sure if PETER is still the SOTA model in 2023 – there are a few follow-up works (listed below) of PETER, which I guess are supposed to be superior to PETER.
>
>
> We will revise the claims accordingly. At the same time, it is worth noting that our goal in this paper is to develop a general framework for improving fairness that can be applied to most of existing explanation generation models, as long as the model is based on user/item representation learning and can be fine-tuned. To that end, we applied COFFEE to both a transformer-based model PETER and RNN-based model NRT (Table 8 in appendix) to demonstrate its generality and effectiveness for different types of models. COFFEE can also be applied to those newer models referenced by the reviewer, since they also depend on learned representations and can be fine-tuned.
>
>
> **Regarding the detailed bias results on Yelp**
> >Line 460: Are there any experimental results that can support this claim?
>
>
> Yes, below are the detailed results supporting our claim in lines 460-464 and we will include them in the appendix. In particular, ground-truth explanations extracted from user reviews show the bias that users generally write detailed and more informative explanations with high FeatCov. When we train a raw PETER model on such data, it also generates higher FeatCov explanations for pricer restaurants than cheaper ones.
> ### Average FeatCov of explanations generated for restaurants of different price tags
> | | $ | $$ | $$$ |
> |--------|----------|---------|---------|
> | Ground-Truth | 3.15 | 3.72 | 4.38 |
> | RAW PETER | 3.26 | 3.68 | 4.03 |
>
>
> **Regarding the correlation to human evaluation after adding Q_{numOfToken}**
> >Given that Figure 2 shows a strong positive correlation between FeatCov and human-evaluated quality measures, after adding Q_{numOfToken}, have authors verified the existence of that correlation – does it remain strong or become weak?
>
>
> Thanks for the question. We confirmed that after adding Q_{numOfToken}, the correlation to human evaluated quality measures remains strong. Similar to Table 5 in appendix D, the table below shows the overall correlation results from Kruskal-Wallis H test between human evaluation and the automatic metric of Q_{sum} = FeatCov + Q_{numOfToken}. All H values are greater than 1 and p-values are much smaller than 0.05 in the test, indicating the significance of the correlations.
>
>
> ### Kruskal-Wallis H test between Q_{sum} = Q_{feat} + Q_{numOfToken} and human evaluated quality measures
> | K-W H test |   | Infor        | Detail       | Help         |
> |------------|---------|--------------|--------------|--------------|
> | $H$ stats | $Q_{sum}$ | 19.16        | 29.47        | 13.87       |
> | $p$-value | $Q_{sum}$ | $7.3e^{-5}$  | $6.26e^{-6}$ | $7.7e^{-4}$ |
>
>
>
> **Regarding the presentation**
>
> Thank you for pointing out the issues regarding our presentation. COFFEE is designed to be a general framework that applies to different personalized text generation (PTG) settings. Focusing on explanation generation does not impose any application-driven assumptions or specific adaptations of COFFEE specific to this setting. We only experimented with explanation generation settings mainly because it is a typical scenario for personalized text generation and widely studied with various datasets and models to experiment with. We will carefully re-examine the scope and revise the paper accordingly.

---

### Meta-Review · Area_Chair_Qte6 · 2023-09-16

**Recommendation:** 4

**Metareview:**

The paper examines the issue of potential bias and unfairness in personalized text generation models. Specifically, it looks at personalized explanation generation for recommendations. The authors find that biases in the user-written training data can lead models to associate different levels of linguistic quality with different user attributes. This results in unfair treatment when serving users with different protected attributes. To address this problem, the paper proposes a new framework called COFFEE to optimize counterfactual fairness constraints during training. The key ideas are disentangling user attribute representations and using policy learning to impose fairness rewards.

The proposed framework COFFEE shows promise in mitigating quality disparities for personalized explanation generation. In particular, experiments on explanation generation datasets demonstrate the system's ability to reduce quality disparities across user attributes.
Looking at the reviews, the reviewers generally acknowledge the importance of studying counterfactual fairness for personalized text generation, particularly in the area of explanation systems. They find the technical contributions of disentangled representations and policy learning to be novel and promising. However, concerns exist around justifying quality metrics as proxies for human preferences and quantifying the harmfulness of observed disparities. The reviewers consistently seek more evidence that differential quality is inherently problematic rather than simply representing user preference differences. Additionally, two reviewers raise doubts about assumptions in the counterfactual inference module, suggesting sensitive attributes may not be fully disentangled from representations by removing them from the input. The framing also overstates the generalizability of the approach beyond the specific explanation generation setting based on the experiments shown. While the reviewers are in agreement that the paper is technically sound overall, there is a general recommendation for authors to enhance the problem formulation, making explicit the required assumptions, and situating the work within the broader literature on algorithmic fairness. In my opinion, the authors have provided satisfactory clarification on most technical concerns, which equally address my own worries about the work. The authors also demonstrated a sufficient understanding of general fairness literature and methodology. I think the technical approach appears novel and promising.

In summary, the paper tackles an important emerging problem at the intersection of fairness, personalization, and natural language generation. The initial results are positive, but further analysis of the assumptions and framework generalizability would strengthen the contribution.

---

### Decision · Program_Chairs · 2023-10-07

**Decision:**

Accept-Main

**Comment:**

The paper examines the issue of potential bias and unfairness in personalized text generation models. Specifically, it looks at personalized explanation generation for recommendations. The authors find that biases in the user-written training data can lead models to associate different levels of linguistic quality with different user attributes. This results in unfair treatment when serving users with different protected attributes. To address this problem, the paper proposes a new framework called COFFEE to optimize counterfactual fairness constraints during training. The key ideas are disentangling user attribute representations and using policy learning to impose fairness rewards.

The proposed framework COFFEE shows promise in mitigating quality disparities for personalized explanation generation. In particular, experiments on explanation generation datasets demonstrate the system's ability to reduce quality disparities across user attributes.
Looking at the reviews, the reviewers generally acknowledge the importance of studying counterfactual fairness for personalized text generation, particularly in the area of explanation systems. They find the technical contributions of disentangled representations and policy learning to be novel and promising. However, concerns exist around justifying quality metrics as proxies for human preferences and quantifying the harmfulness of observed disparities. The reviewers consistently seek more evidence that differential quality is inherently problematic rather than simply representing user preference differences. Additionally, two reviewers raise doubts about assumptions in the counterfactual inference module, suggesting sensitive attributes may not be fully disentangled from representations by removing them from the input. The framing also overstates the generalizability of the approach beyond the specific explanation generation setting based on the experiments shown. While the reviewers are in agreement that the paper is technically sound overall, there is a general recommendation for authors to enhance the problem formulation, making explicit the required assumptions, and situating the work within the broader literature on algorithmic fairness. In my opinion, the authors have provided satisfactory clarification on most technical concerns, which equally address my own worries about the work. The authors also demonstrated a sufficient understanding of general fairness literature and methodology. I think the technical approach appears novel and promising.

In summary, the paper tackles an important emerging problem at the intersection of fairness, personalization, and natural language generation. The initial results are positive, but further analysis of the assumptions and framework generalizability would strengthen the contribution.